# An Augmentation a Day Keeps the MRI Away: Blueprint Augmentation for Brain Graphs

Anonymous Author[1], Anonymous Author[2], and Anonymous Author[3]

[1]Anonymous Institution 1
[2]Anonymous Institution 2
[3]Anonymous Institution 2

**Abstract.** Graph augmentations are essential for expanding datasets for classification tasks. Existing graph augmentation techniques often fail to preserve the graph's topological structure. This is particularly important for brain graphs were changes in topology could lead to graph misclassification (i.e., misdiagnosis of brain conditions). This paper introduces Blueprint Graph Augmentation (BluGrAu), a proof-of-concept method designed to address the challenge of augmenting brain graphs derived from MRI data without compromising topological integrity. BluGrAu leverages the novel concept of a "graph blueprint," which is a *transformed* version of an input graph where *essential* topological features of an input graph are preserved. The graph blueprint aims to serve as a template for generating graph variations that retain critical structural properties. By employing a graph neural network-based variational autoencoder (VAE), BluGrAu produces augmented graphs that improve classification accuracy while maintaining a consistent topological structure.

**Keywords:** Brain graph augmentation, Topology-aware augmentation, Graph Variational Autoencoder

## 1 Introduction

Brain graphs, often derived from Magnetic Resonance Images (MRIs), provide a representation of structural or functional connectivity between different regions of the brain. These graphs could be used for diagnosing neurological disorders such as Alzheimer's disease and Autism [3]. Recent studies demonstrate that Graph Neural Networks (GNNs) can utilize brain graph data to distinguish between healthy and diseased states [7]. However, GNN-based methods face challenges when limited training data are available, as robust classification typically requires larger datasets [8]. Although generating additional brain graphs by augmenting MRI scans can mitigate data scarcity, it comes with the risk of bias where minor pixel-level changes in an MRI can produce different graphs, potentially altering final diagnosis.

Graph augmentation techniques have been investigated to expand dataset sizes. For example, [9] introduces a method for random edge removal to create augmented graphs for classification tasks. This method treats all edges equally, hence potentially disregarding crucial connections. To enable effective graph augmentation while maintaining topological integrity, [2] introduced a topology-aware augmentation technique called *Topology Adaptive Edge Dropping* (TADropEdge). This method utilizes a GNN to selectively remove edges based on their topological relevance and influence within a given dataset. While TADropEdge effectively augments input graphs and preserves their topology, any edge removal, regardless of the edge relevance, leads to distortion of the graph topology. Additionally, relying on removing only certain *irrelevant* edges, limits the number of augmented graphs to the number of possible combinations of edges to be removed. In this work, we aim to answer the following question: *How can we augment brain graphs while preserving all edges and maintaining the overall topology?*

To address this, we draw from the analogy of image augmentation, where transformations like recoloring or scaling alter an image's appearance without changing its core structure. Similarly, we seek transformations—such as edge weight adjustments—that preserve topology. We introduce *Blueprint Graph Augmentation (BluGrAu)*, a GNN-based method for generating *unlimited* augmented graphs while retaining the initial topological structure. BluGrAu aims to enhance classification by producing graph variations via topology-preserving augmentations. We define a *graph blueprint* as a transformation of a given input graph into a fully-connected graph where each node feature vector represents topological attributes, such as betweenness centrality, and edges denote pairwise node feature similarities.

Our proposed **BluGrAu** aims to augment a given input graph by training a model that maps a combination of a blueprint graph and white noise to its target graph using a GNN. We assume that generated graphs are variations from the target graph while preserving input graph topology. Our contributions are as follows: (i) We introduce the concept of a **graph blueprint**, which acts as a topological representation that compacts the structural information of the graph in a separate graph called graph blueprint. (ii) We demonstrate a proof of concept for using graph blueprints with GNNs to generate augmented brain graphs that are used to improve classification tasks.

## 2   Method

### 2.1   Concept: Graph Blueprint

Let $\mathcal{G}(V, E)$ represent a brain graph, where $V$ denotes brain regions as nodes where each node features is a vector $C$ containing the connectivity weights of that node to every other node in the graph, and $E$ represents weighted edges indicating pairwise connection strength between nodes. A graph blueprint $\mathcal{G}_b(V_b, E_b)$ simplifies $\mathcal{G}$ to serve as its structural template. In particular, as detailed below

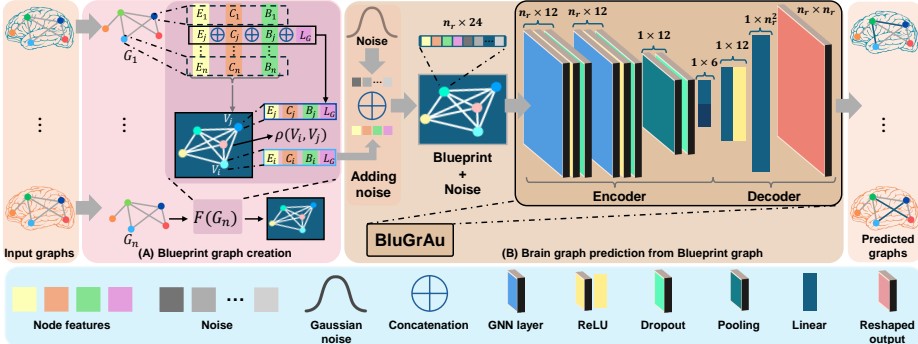

**Fig. 1.** *BluGrAu* **Architecture Overview.** (A) Blueprint graph creation: For each node in an input brain graph, we calculate the betweenness, closeness, and eigenvector centralities. These, combined with the graph label, form the blueprint graph's node features. (B) BluGrAu training pipeline: we concatenate each blueprint's graph node with a noise vector, the obtained new blueprint+noise is processed by a graph VAE to predict target brain graphs. The VAE comprises an encoder with two GNN layers followed by ReLU and dropout, then global pooling and dropout. The decoder includes two linear layers, with output reshaped to match the input graph's dimensions. We denote by $G_{1..n}$ the input graphs, $E_{1..n}, C_{1..n}, B_{1..n}$ the eigenvector, closeness, and betweenness centrality measures, respectively, $\rho(V_i, V_j)$ the Pearson correlation between nodes' features of $V_i$ and $V_j$, by $L_G$ the label of the graph, and by $F(G_n)$ the function that generates a blueprint graph from an input graph.

in Algorithm 1, each node $V_b^i$ captures three key topological measures: betweenness, closeness, and eigenvector centralities as well as the input graph label. Despite the existence of several graph centrality measures (e.g., Katz centrality), these measures were chosen for their ability to jointly capture minimal core roles within the network. For instance, betweenness centrality identifies nodes bridging distant regions, while closeness centrality reflects a node's accessibility across the network, and eigenvector centrality highlights influence through connections with other important nodes. Together, they provide a minimal profile of topological information for each node. To complete the graph structure of $\mathcal{G}_b$, each edge $E_b$ represents the pairwise Pearson correlation between feature sets associated with a given pair of nodes $V_b^i$ and $V_b^j$.

## 2.2   BluGrAu for Augmented Brain Graph Generation

The *BluGrAu* model is a Graph Variational Autoencoder (VAE) designed to generate topology-aware graph augmentations from input brain graphs $\mathcal{G}(V, E)$. We utilize a GNN-based encoder, where $\mathcal{G}_b(V_b, E_b)$ represents the input blueprint graph. Fig. 1 shows the architecture comprising of an encoder and a decoder which are detailed as follows:

**Encoder.** The encoder is implemented using a two-layer GNN, where each layer is followed by non-linear activation and dropout regularization. Initially,

node features from $\mathcal{G}_b$, with dimensions $N \times I_f$ ($N$ is the number of nodes and $I_f$ is the feature dimension per node), are concatenated with random noise features (set to 20) and then fed into the first GNN layer. The added noise aims to prevent overfitting and enhance model robustness to noisy node features. The first layer projects the node features into a hidden space of dimensions $N \times 12$. The second GNN layer then processes this $N \times 12$ output while preserving the hidden dimension of 12.

To introduce stochasticity in the latent space, the encoder performs a global mean pooling operation across node embeddings which aggregates them into a graph-level embedding of size $1 \times 12$. This pooled embedding is then mapped through two fully connected layers to produce the mean ($\mu$) and log-variance ($\log \sigma^2$) vectors, parameterizing a Gaussian distribution specific to each graph in the batch. These vectors, of dimension $1 \times 6$, enable the reparameterization trick, where the latent vector $z$, which serves as the final encoded representation, is sampled as $z = \mu + \epsilon \cdot \sigma$, with $\epsilon \sim \mathcal{N}(0, I)$ and $\sigma = \exp(0.5 \cdot \log \sigma^2)$. This ensures that the latent representations are differentiable to allow backpropagation through stochastic nodes.

---

**Algorithm 1** Generate Blueprint Graphs

---

**Require:** Directed weighted graph $\mathcal{G} = (V, E)$
**Ensure:** Blueprint graphs $\mathcal{G}_b$
1: **for** node $i$ in $\mathcal{G}$ **do**
2:      Initialize $\mathcal{G}_b \leftarrow \emptyset$
3:      Calculate betweenness, closeness, and eigenvector
4:      Concatenate values into node feature vector
     $V_b^i \leftarrow$ [eigenvector, closeness, betweenness, graph label]
5: **end for**
6: **for** node $i$ in $\mathcal{G}_b$ **do**
7:      **for** node $j$ in $\mathcal{G}_b$ **do**
8:          $E_b^{i,j} \leftarrow \text{Pearson}(E_b^i, E_b^j)$
9:      **end for**
10: **end for**
11: **return** $\mathcal{G}_b$

---

**Decoder.** It comprises two linear layers to reconstruct the adjacency matrix $\hat{\mathcal{G}}$ from the latent representation $z$. The first linear layer maps from the latent dimension 6 to a hidden dimension of 64, followed by a ReLU activation. The second layer reshapes the output to a tensor of size $(90 \times 90)$, corresponding to the flattened adjacency matrix format.

**Training Procedure.** The *BluGrAu* model is trained for each class (i.e., graph label) independently over 100 epochs using Adam optimizer [5] with a learning rate of 0.005. We employ an L1 loss to compare the reconstructed adjacency matrix $\hat{\mathcal{G}}$ to the ground truth matrix $\mathcal{G}$. Additionally, a KL divergence term is included to regularize the latent space distribution, calculated as:$\text{KL} = -0.5 \cdot \sum (1 + \log(\sigma^2) - \mu^2 - \sigma^2)$ The total loss function is defined as:

$\mathcal{L} = \mathcal{L}_1 + \beta \cdot \mathrm{KL}$, where $\beta$ is a weighting parameter empirically set to control the contribution of the KL divergence.

**Implementation Details.** The model was implemented using the PyTorch Geometric library [1]. The training was conducted on Google Colab using its available CPUs. The hidden dimensions and dropout rates were empirically chosen to prevent overfitting while allowing model complexity to capture graph structures. Code is available in this link: (**shared upon acceptance**.)

## 3    Results and Discussion

**Dataset and Preprocessing.** We evaluated *BluGrAu* using a dataset described in [10], comprising 88, healthy individuals (48 females and 40 males, ages 18 to 48). Each subject's structural connectome includes 90 brain regions, defined by the Automated Anatomical Labeling Atlas (AALA) [12]. The dataset was preprocessed following the procedures outlined in [10], which include quality checking and artifact removal, distortion correction, skull stripping, spatial registration of T1 and DWI images to the MNI template, generation of white matter masks, and probabilistic tractography to construct normalized connectivity matrices based on streamline counts between AAL [12] atlas ROIs.

### 3.1    Quantitative Evaluation of Predicted Graphs

**Table 1.** Evaluation table

| Method | MAE ↓ | F1-score ↑ | GRAM ↓ |
|---|---|---|---|
| Without Augmentation | N/A | 0.7 | - |
| BluGrAu w/ GCN | 0.0054±0.01 | 1.0 | 8.34% |
| BluGrAu w/ GAT | 0.0052±0.01 | 1.0 | 19.13% |
| BluGrAu w/ GraphSAGE | 0.0050±0.01 | 1.0 | 9.72% |

We train our VAE with 3 distinct GNN configurations: A graph convolutional network (GCN) [6], a graph attention network (GAT) [13] and GraphSAGE [4]. We split data into 80% training and 20% testing. Each testing graph is augmented 100 times. Table 1 shows the Mean Absolute Error (MAE) as well as the classification F1 score (males vs females) using a 5-layer Multi-layer Perceptron (MLP). The MLP inputs flattened brain graphs into vectors of size $8100 \times 1$ and outputs vectors of the following sizes at the output of each layer: 512, 256, 64, 8, and 1, respectively. The MLP is trained over 75 epochs and with a learning rate of 0.01 and Adam optimizer [5]. Additionally, we employ Graph Regularizable Assessment Metric (GRAM), which aims to linearly measure the distortion between the ground-truth and augmented graphs using a selection of 10 graph topological metrics as in [11].

Table 1 presents the model's performance metrics, indicating that the MAE is low, reflecting strong alignment between predicted and reference graphs. Furthermore, the F1 score—encompassing both precision and recall—rose significantly from 0.7 for non-augmented test graphs to 1.0 with augmentation. Additionally, the similar performance across all three GNN layer configurations indicates that the choice of aggregation function in each graph convolutional layer has minimal impact on model accuracy in this setup, suggesting that the augmentation itself is the primary contributor to the improved results. As for GRAM, we notice that our predicted graphs using BluGrAu trained with GCN scored 8.34%, the lowest topological deviation from the target graphs, while the predicted graphs using BluGrAu and GAT scored the highest topological deviation at 19.13% from the target graphs.

### 3.2   Qualitative Evaluation of Predicted Graphs

Fig. 2 displays the reference, prediction, and absolute difference for an exemplar brain graph using the abovementioned 3 GNN configurations in the VAE. Visually, the difference seems to be markedly minimal between reference and predicted graphs. This indicates that *BluGrAu* can be used to predict graphs that closely match the reference graphs. Additionally, by visual inspection, we notice that most highly-weighted connectivities are close to the diagonal in the reference graphs, a close connectivity scheme is also visible in the predicted graphs which shows that our model is able to preserve a topological consistency with the reference graphs.

## 4   Conclusion and Future Work

We introduced *BluGrAu*, a topology-preserving augmentation proof-of-concept method for brain graphs, using the graph blueprint framework within a graph VAE architecture. Our results demonstrate that *BluGrAu* effectively generates augmented graphs that maintain the topological structure of input graphs and improve classification performance. Future work will extend this approach to several brain graph datasets and qualitative assessments to rigorously quantify topological preservation.

## 5   Compliance with Ethical Standards

No ethical approval was required.

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
