# OpenReview forum: "An Augmentation a Day Keeps the MRI Away: Blueprint Augmentation for Brain Graphs"
_MICCAI.org/2025/Workshop/MSB_EMERGE — Submitted to MSB EMERGE 2025_

### Official Review · Reviewer_1Dks · 2025-07-01

**Recommendation:** 2
**Confidence:** 2

**Clarity:**

The paper is unclear and difficult to understand due to significant clarity issues, major revision is necessary

**Feedback:**

- The paper is hard to read as each section is a bit confusing. E.g. in Sec. 3.1 you start by referencing the table and explaining MAE, and F1 score, followed by an explanation of an MLP and suddenly the explanation of GRAM. Instead, you could dedicate separate subsections or paragraphs to the metrics, the MLP, and the results. The entire paper suffers from this issue.
- I dislike citing with numbers within text, e.g. "[2] introduced". Instead, use \citet and \citep which will create better readable text!
- The running title name is too long!
- An introductory Figure might look good and help with understanding the brain graphs in general
- floats are placed very oddly. For example, Fig. 2 comes after the first reference, which is a no-go. I barely even noticed it when first scanning the paper.
- As I said before, the manuscript is pretty confusing. Especially, if you have a lot of space, use subsections, paragraphs etc. to structure your paper.
- You must write a lot more. For example, the caption of Table 1 is useless.
- For the algorithm, I'd recommend using comments and math symbols.
- Fig. 1 looks awesome. When describing the method, adhere to the Fig: First, describe the blueprint graph creation, including Alg. 1. Then, follow with the brain graph prediction. Maybe write a simple algorithm for it as well.
- In your case, it is fine as you have enough space, but generally speaking, you are wasting a lot of space with implementation details that are too detailed for your method.
- In your abstract place BluGrAu in \mbox{} so it does not break lines

**Justification:**

Presentation, motivation and results are mediocre. Hence, I would recommend rejecting this paper.

**Reproducibility:**

Sufficient amount of details available for reproducing the main results, and open access is provided (or promised upon acceptance) to source code and/or data

**Strengths:**

- The title is funny.
- Figure 1 is amazing. It not only looks good, but also clearly explains the methodology, supplementing the text well.
- The augmentations from BluGrAu help in training a classification model

**Summary:**

The authors propose a method for augmenting brain graphs while preserving the topology of the graph by computing new edge weights using a VAE.

**Weaknesses:**

- The paper is hard to read as each section is a bit confusing.
- The manuscript misses an introduction to brain graphs. What are they, how are they acquired, how are they represented, and what are they used for?
- Due to the missing introduction, I do not really see the sense in the motivation. Why would you base your prediction on a brain graph, rather than an MRI?
- The proposed method leaves the nodes and the associated features invariant. Only the edges specifying the "pairwise connection" are changed. Why does that make sense? I mean the graph stays exactly the same, so why should edge weights change?
- How much do different graphs vary? I am surprised that a latent dimension of only $6$ is enough to compute reasonable outputs of dimensionality $90\times90=8100$.
- The paper misses baselines. For example, adding a relatively small amount of noise to each feature without the need for a trained model.
- I believe in Alg. 1, line 2 must be _outside_ of the for loop!

---

### Official Review · Reviewer_WtRd · 2025-07-08

**Recommendation:** 2
**Confidence:** 3

**Clarity:**

The paper has significant clarity issues that hinder understanding, substantial revision is required to improve clarity

**Feedback:**

- Add Disease Classification Experiments: Evaluate the effect of augmented graphs on brain disease classification (e.g., Alzheimer’s vs. controls) to demonstrate clinical value.

- Expand Topology Metrics: Incorporate or compare additional centrality measures (Katz, PageRank) in the blueprint to identify optimal feature combinations.

- Enhance Decoder Complexity: Explore nonlinear or GNN-based decoder architectures to improve fidelity of reconstructed connectivity.

- Scalability Testing: Conduct experiments on GPU and larger brain graphs (e.g., >200 ROIs) to analyze speed and memory usage.

- Expert Qualitative Evaluation: Include expert assessment of augmented graphs for preservation of meaningful brain networks (e.g., DMN, synaptic pathways).

**Justification:**

BluGrAu addresses a critical challenge in graph augmentation by introducing blueprint-driven, topology-preserving generation. It validates the method across multiple GNN backbones with robust metrics. However, demonstrating clinical applicability, expanding blueprint features, and improving decoder complexity are necessary to strengthen the work.

**Reproducibility:**

Sufficient amount of details available for reproducing the main results, and open access is provided (or promised upon acceptance) to source code and/or data

**Strengths:**

- Topology-Preserving Augmentation: Generates augmented graphs without adding or removing edges by leveraging blueprint-based VAE, effectively maintaining global graph structure.

- Unlimited Sample Generation: Gaussian noise input enables generation of an arbitrary number of augmented graphs, alleviating data scarcity.

- Backbone Compatibility: Demonstrates consistent performance gains across diverse GNN encoders (GCN, GAT, GraphSAGE), highlighting modularity and generality.

- Quantitative and Qualitative Evaluation: Shows improvements in MAE and F1, uses GRAM to quantify topology distortion, and provides visualizations (Fig.2) to confirm structure preservation.

**Summary:**

This paper proposes Blueprint Graph Augmentation (BluGrAu) for topology-preserving augmentation of brain graphs that represent structural/functional connectivity. It first generates a “graph blueprint” by combining key topological features (betweenness, closeness, eigenvector centrality) and graph labels. Then, Gaussian noise is added to the blueprint and fed into a graph variational autoencoder (VAE). The decoder produces augmented graphs that preserve the original connectivity structure. Experiments on a structural connectivity dataset of 90 brain regions (AAL-90) from 88 healthy subjects, evaluated with GCN, GAT, and GraphSAGE encoders, demonstrate improvements in MAE, F1 score, and GRAM (graph topology distortion metric) .

**Weaknesses:**

- No Clinical Validation: Only healthy subject data are used; the impact on disease classification (e.g., Alzheimer’s, autism) is not assessed.

- Limited Blueprint Features: Relies solely on three centrality measures without comparing other topology metrics (e.g., Katz centrality).

- VAE Decoder Simplicity: Uses a linear decoder, raising concerns about its capacity to reconstruct complex brain connectivity patterns accurately.

---

### Official Review · Reviewer_3Ztv · 2025-07-09

**Recommendation:** 3
**Confidence:** 3

**Clarity:**

The paper has significant clarity issues that hinder understanding, substantial revision is required to improve clarity

**Feedback:**

1. Instead of merely stating that visual inspection of the qualitative results is favorable, it would be more convincing to include additional figures to support this claim.
2. Add experiment results using diverse datasets (datasets that have diseases).
3. Please include comparative experiments between MRI-level augmentation and your proposed graph-level augmentation to better demonstrate the effectiveness of your approach.
4. Explain why betweenness, closeness, and eigenvector centralities were chosen as topological measures, given the existence of other centrality measures such as Katz centrality. There are no theoretical or experimental evidence that supports this choice.

**Justification:**

The paper emphasizes the ability to generate unlimited augmented brain graphs without edge deletions, preserving essential structural integrity. However, several modifications, e.g., addition of experimental results on diverse datasets, should be made for the paper to be accepted.

**Reproducibility:**

Some amount of details available for reproducing the main results, and open access details are unclear

**Strengths:**

1. Objective of the study is clearly represented in Section 1.
2. Representation of Fig. 1 is very clear.
3. The paper introduces a novel concept of a “graph blueprint” that preserves key topological features of brain graphs, which is particularly relevant for medical applications where topological consistency is critical.

**Summary:**

This paper introduces BluGrAu (Blueprint Graph Augmentation), a novel topology-preserving graph augmentation method designed specifically for brain graphs derived from MRI data. The method addresses a key limitation in current graph augmentation approaches that the distortion of topological structures that can critically affect medical graph-based diagnoses. BluGrAu proposes two key contributions:  Blueprint Graph Construction – where a given brain graph is transformed into a “blueprint” graph using node-level centrality metrics (betweenness, closeness, eigenvector) and edge-wise Pearson correlations.   Augmented Graph Generation – a graph variational autoencoder (VAE) takes blueprint+noise as input and reconstructs brain graphs that preserve original topology while introducing realistic variations. Experiments are conducted on a dataset of 88 healthy individuals' structural connectomes for validation.

**Weaknesses:**

1. While the proposed method shows promising topological consistency, the clinical validity of the augmented graphs remains unclear, as no experiments were conducted on disease classification tasks or with pathological datasets. Further validation on clinically meaningful use cases is required to support the claim that augmented graphs are medically reliable.
2. Although the method is introduced as a solution to reduce the risk of misdiagnosis by augmenting brain graphs, the dataset used consists solely of healthy individuals. It is unclear what meaningful classification tasks can be achieved under this setting.
3. While the paper stresses that the proposed method can be used on behalf of real MRI images (as mentioned in the title), comparisons are not made to validate this statement.
4. Explanations on why certain methods were chosen are not clearly detailed.